# Growth Potential of *Listeria monocytogenes* in Three Different Salmon Products

**DOI:** 10.3390/foods9081048

**Published:** 2020-08-03

**Authors:** Corinne Eicher, Andres Ruiz Subira, Sabrina Corti, Arnulf Meusburger, Roger Stephan, Claudia Guldimann

**Affiliations:** 1Institute for Food Safety and Hygiene, Vetsuisse Faculty, University of Zürich, 8057 Zürich, Switzerland; corinne.eicher@uzh.ch (C.E.); andres.ruizsubira@uzh.ch (A.R.S.); cortis@fsafety.uzh.ch (S.C.); stephanr@fsafety.uzh.ch (R.S.); 2Ospelt Food AG, 7320 Sargans, Switzerland; Arnulf.Meusburger@ospelt.com

**Keywords:** *Listeria monocytogenes*, cold smoked salmon, food safety, sushi, sodium lactate, growth potential

## Abstract

Cold smoked salmon and sushi salmon have been implicated in outbreaks of listeriosis. We performed challenge tests and a durability study with *Listeria monocytogenes* on different salmon products to determine the growth potential of this important food-borne pathogen. Data from the challenge test showed a significant growth potential of *L. monocytogenes* on all of the tested salmon products, with faster growth in sushi salmon than in cold smoked salmon. In identical products that were naturally contaminated at low levels, the durability study did not confirm a high growth potential, possibly due to interactions with competing microflora. The injection of sodium lactate (NaL) at a high concentration (30%) into cold smoked salmon significantly reduced the growth potential of *L. monocytogenes*. In addition to good manufacturing practices, the injection of higher concentrations of NaL may therefore be a useful additional hurdle to prevent growth of *L. monocytogenes* to high numbers in the tested salmon products.

## 1. Introduction

*Listeria monocytogenes* is the causative agent of listeriosis. In particular the elderly, pregnant and immunosuppressed people are at increased risk for listeriosis [1]. Although the number of cases is low [2], the consequences of clinical listeriosis can be severe due to its ability to cause septicemia, abortions and infections of the central nervous system [3].

In most cases, listeriosis is acquired through consumption of contaminated food products. *Listeria monocytogenes* is able to grow in high salt concentrations (7–9% wt/vol) [4], at cold storage temperatures (4 °C) [2,5], and to survive under the selective pressure of commonly used disinfectants such as benzalkonium chloride and peracetic acid [6]. These properties make *L. monocytogenes* prone to persist in food production facilities. The complex task of keeping food production environments free of *L. monocytogenes* combined with the severity of the disease with mortality rates between 15–30% [7], make *L. monocytogenes* a food safety priority for food business operators (FBO) as well as the regulatory bodies.

Among food categories, ready-to-eat (RTE) foods pose a high risk for transmission of *L. monocytogenes* to consumers due to the absence of a microbial kill step such as cooking prior to consumption. In the EU, 17.4% of 599 of cold smoked fish samples were positive for *L. monocytogenes* in the EU baseline survey conducted in 2010/2011 [8]. While the numbers of *L. monocytogenes* in these products were generally below 1 log CFU/g [9,10,11], reports of contamination of cold smoked fish products at high levels of 5–6 log CFU per g are available [12,13]. In recent years in the EU, several outbreaks involved RTE meats [14,15] and seafood [16,17,18,19]. These events demonstrate the challenges associated with the control of *L. monocytogenes* in these food categories because their intrinsic factors, e.g., high a_w_ values, support growth of *L. monocytogenes*. To reduce the risk by *L. monocytogenes* contamination to consumers, lactic acid and its salts, e.g., sodium lactate (NaL), have been used as antimicrobial compounds in high-risk foods. They are generally recognized as safe (GRAS) by the US Food and Drug Administration and have been used to extend shelf life, enhance flavor and to increase cooking yields in various meat products [20,21,22,23] and fish [24,25].

There is no consensus on acceptable levels of *L. monocytogenes* in food. While the USA enforces a zero-tolerance policy, EU regulations stipulate a food safety criterion of <100 CFU/g during the shelf life of the product (Commission Regulation (EC) No 2073/2005). To achieve this, the EU legislation states different testing criteria depending on whether a product allows the growth of *L. monocytogenes*. RTE foods that do not allow the growth of *L. monocytogenes* are tested to demonstrate levels of *L. monocytogenes* <100 CFU/g in five samples. RTE foods that allow the growth of *L. monocytogenes*, are tested for the absence of *L. monocytogenes* in 5 samples of 25 g (EC regulation No 2073/2005), with even stricter criteria for foods intended for infants and special medical purposes.

It is therefore necessary for FBOs in Europe to assign their RTE products to these categories with confidence. To assess the growth behavior of *L. monocytogenes* in an RTE food product, FBOs may perform challenge tests by artificially inoculating samples with *L. monocytogenes*, conduct durability studies with naturally contaminated samples, or model the growth potential of *L. monocytogenes* in their RTE product based on existing data (EC regulation No 2073/2005). The aim of this study was to comparatively examine the growth potential of *L. monocytogenes* in three different salmon products in a challenge test (artificially contaminated products) and a durability study (naturally contaminated products).

## 2. Materials and Methods

The challenge tests in this study were performed in accordance with the “EURL Lm Technical Guidance Document for conducting shelf-life studies on *Listeria monocytogenes* in ready-to-eat food”, version 3–6 June 2014 [26]. Two minor modifications to the technical guidance document were made. (i) We used *L. monocytogenes* strains that were isolated from a salmon processing facility instead of the reference strains because we deemed them directly relevant. (ii) The technical guidance document states that products shall be inoculated within 2 days of their production date. In accordance with regulation (EC) No 853/2004, the sushi salmon products used in this study were frozen by the producer immediately after production and stored frozen. The other salmon products were stored at −3 °C after packaging. They were then thawed on the day they were inoculated (t = 0).

All three strains that were used in this study were isolated from a salmon processing food facility and characterized by serotyping and whole genome sequencing to determine the core-genome multilocus sequence type (MLST) and traditional seven-gene MLST (Table 1). Bacteria were stored at −80 °C in brain heart infusion (BHI, Oxoid, Basel, Switzerland) with 15% glycerol.

For the preparation of the inoculum, glycerol stocks for all three strains were streaked on a BHI agar plate (Brain Heart Infusion, Oxoid, Basel, Switzerland) and incubated overnight at 37 °C. In the next step, a single colony was used to inoculate 5 mL BHI and incubated for 17–19 h at 37 °C with shaking at 200 rpm. The next day, the cultures (approximately 9 log CFU/mL as determined by plate counting) were diluted in Maximum Recovery Diluent (MRD; Oxoid, Basel, Switzerland) and inoculated into 10 mL of BHI broth with a target concentration of 2 log CFU/mL. The cultures were then incubated at 5 °C without shaking for cold-adaption for 13 days until the strains reached early stationary phase (approximately 8 log CFU/mL as determined by plate counting). All three strains were grown separately. The final inoculum consisted of a pool of equal amounts (0.5 mL) of the cold-adapted strains grown to early stationary phase, diluted in 0.85% NaCl to achieve the target contamination of 2 log CFU/g. The inoculum was used immediately.

To assess the growth characteristics of the three *L. monocytogenes* strains, growth curves were performed under the same storage conditions as the challenge test at 5 °C and 8 °C in BHI. Five °C simulates the production temperature at the factory and the recommended storage temperature of the product. Eight °C was selected as an example for abusive storage conditions. Cold-adapted cultures were prepared as described above and incubated at 5 °C and 8 °C. Growth was determined by direct plate counting every day until t = 15 days. Plate count results were expressed in log CFU/mL. Growth curves were done in triplicate.

For this study, salmon products (*Salmo salar*) from aquaculture were used. None of the products were treated with phosphate. The characteristics of the samples used in the present work are presented in Table 2. Norwegian smoked salmon and salmon fillets were delivered to our laboratory one day before the start of the challenge test, in a cooler to maintain the cold chain at 5 °C. The salmon was then stored at 4 °C before use. The sushi salmon was delivered frozen 4 days before the start of the challenge test. This salmon product was frozen at −20 °C and thawed at 4 °C overnight before use.

Based on the assumption that salmon would be supporting *Listeria monocytogenes* growth, all experiments were performed on three different batches for each salmon product as suggested in the EURL Lm Technical Guidance document [26].

Preparation of the samples was the same for all salmon products. All procedures were performed on ice to maintain the cold chain. For each time point 14 samples were produced, seven to be stored at 5 °C and seven at 8 °C, respectively. For each temperature, three units were inoculated with *L. monocytogenes*, three served as negative controls, and to one unit not inoculated with *L. monocytogenes,* NaCl was added for a_w_ and pH measurements. One sample unit consisted of 10 g of the corresponding salmon. For each salmon variety, one additional non-inoculated sample was prepared to detect natural contamination with *L. monocytogenes* at t = 0. The salmon was placed on a 7.5 × 6 cm piece of the coated carton product carrier that was used in the packaging of the commercially sold product. The carton pieces with the salmon were placed in Petri-dishes and the salmon samples were inoculated either with *L. monocytogenes* (2 log CFU/g target concentration) or with sterile 0.85% NaCl for the negative controls. One hundered μL of the inoculum was distributed over the sample with a pipette and then dispersed with the aid of a L-shaped spreader. The inoculated and control samples, including the carton carrier, were placed in polyamide-polyethylene vacuum bags (Solis vacuum packaging bags no 922.61, Glattbrugg, Switzerland) with a pair of sterile tweezers, the bags were vacuumized and sealed with a commercially available vacuum sealer (Solis Easyvac Pro 569, Glattbrugg, Switzerland).

The samples were stored at 5 °C and 8 °C, respectively, for the entire duration of the shelf life (Table 2). The growth of *L. monocytogenes*, total viable counts (TVC) and the physicochemical properties of the samples were determined at the following time points: immediately after inoculation (t = 0) and on day 2 and 3 for the sushi salmon. Norwegian smoked salmon was analyzed on day 0, 5, 12, 13, 14, 15, 16. Salmon fillets were analyzed on day 0, 5, 11, 12, 13, 14, 15. At t = 0, the additional, uninoculated sample for each salmon type was analyzed to verify the absence of *L. monocytogenes* in the raw material (at the limit of detection of 1 log CFU/g). These time points were chosen to reflect the use-by date (corresponding to d = 3 for sushi salmon, d = 16 for Norwegian smoked salmon and d = 15 for salmon fillets) given for each salmon type by the producing company.

The physiochemical properties of the samples were determined. At each time point, pH and a_w_ values were determined for one non-inoculated bag stored at 5 °C and 8 °C, respectively. An Aqualab (Series 3TE, Meter Group, USA) was used to measure the a_w_ value and a solid matter pH meter (Mettler Toledo, Greifensee, Switzerland) was used to determine the pH.

In addition to the enumeration of *L. monocytogenes*, total viable counts (TVC) for the salmon- associated microflora were determined in the samples. TVC was not only determined in non-inoculated samples, as mandated by the EURL Lm Technical Guidance document [26], but also in inoculated samples to determine the behavior of the microflora in the presence of *L. monocytogenes*. At each time point, 90 g of MRD (corresponding to a 1:10 dilution) was added to three inoculated and three control samples. After homogenization with a Stomacher^®^ (400 Circulator, Seward, Worthing, UK) for 30 s at medium speed, serial dilutions were made in 10 mL of MRD. Enumeration of *L. monocytogenes* was performed by pour-plating in Palcam Agar in triplicate, with a limit of detection of 1 log CFU/g. TVC was determined by spread-plating 100 µL of the serial dilutions on PC-plates (Plate count; Oxoid, Basel, Switzerland) in duplicates. Palcam plates were incubated at 37 °C for 48 h and PC plates at 30 °C for 72 h, respectively, before colonies were counted.

The growth potential δ was determined according to the EURL Guidance document as “the difference between the median of results at the end of the challenge test and the median of results at the beginning of the challenge test” [26]. The final growth potential was given as the highest value among the three replicates for each salmon product.

The durability study was also performed according to the EURL Guidance document. In contrast to a challenge test, where samples are artificially inoculated with a pathogen of interest, a durability study observes the growth of a pathogen in naturally contaminated products [26]. The durability study was performed for 10 days in sushi salmon, for 14 days in salmon fillet and for 16 days in Norwegian smoked salmon. At each time point, five samples of a product were tested for the presence of *L. monocytogenes*. For quantitative determination of *L. monocytogenes*, 25 g of salmon was diluted 1:10 in MRD of which 1 mL was spread-plated on OCLA agar (Oxoid, Pratteln, Switzerland), resulting in a limit of detection of 1 log CFU/g. Qualitative detection of *L. monocytogenes* was done by enrichment in Half-Frazer broth (Oxoid, Pratteln, Switzerland) for 48 h followed by real-time PCR in a GDS system (Merck, Basel, Switzerland).

Statistical analyses were performed in R (version 3.6.2), using Knime (version 4.1.0) and RegressIT (version 2020.2.23); and SPSS (version 25). Data visualization was done in R (Version 3.6.2) using R studio (Version 1.2.5033) and the ggplot2 package (Version 3.2.1) [27]. Data quality analysis and outlier detection was performed with Knime, using a nonparametric outlier detection method in a one-dimensional feature space. Outliers were calculated by means of the IQR (InterQuartile Range). The first and the third quartile (Q1, Q3) were calculated and outliers defined as datapoints xi that were outside the interquartile range. Therefore: xi > Q3 + k (IQR) ∨ xi < Q1 − k (IQR), where IQR = Q3 − Q1 and k ≥ 0. Using the interquartile multiplier value k = 1.5, the range limits are the typical upper and lower whiskers of a box plot. Linear regressions were conducted with Regressit (Version 2020.02.23), to calculate the correlation between different series with a confidence level of 0.95.

“Effect size” [28] was used to determine statistical significance between two different series, *p*-values < 0.05 were considered significant. The T score was calculated with SPSS.

## 3. Results

### 3.1. Growth Curve of the Three L. monocytogenes Strains in Non-Selective Growth Medium and Physiochemical Characteristics of the Samples

Growth at cold temperatures was identical for the three *L. monocytogenes* strains used in this study (Appendix A). All strains reached early stationary phase after 13 days at 5 °C and after 6 days at 8 °C, respectively.

The salmon products used in this challenge test had a_w_ values ranging from 0.96–0.99 and a pH of 6.0–6.1 (Table 2).

### 3.2. Growth Characteristics of L. monocytogenes in the Different Salmon Varieties

Starting with a target inoculation of 2 log CFU/g, *L. monocytogenes* numbers significantly increased in all salmon products (*p* < 0.05 at all time points compared to t = 0), both at 5 °C and 8 °C (Figure 1). The highest numbers of *L. monocytogenes* were observed in salmon fillets at day 15 with 4.6 ± 1.14 log CFU/g at 5 °C and 5.5 ± 0.75 log CFU/g at 8 °C, respectively.

Due to the short shelf life of the sushi salmon, growth of *L. monocytogenes* was only investigated between t = 0 and t = 3 days. Over this period, the number of *L. monocytogenes* increased by 1.0 log CFU/g at 5 °C and 1.7 log CFU/g at 8 °C, respectively. The slope of the growth curves of *L. monocytogenes* in sushi salmon (0.33 at 5 °C, 0.57 at 8 °C) were steeper than those in Norwegian smoked salmon (low NaL) (0.05 at 5 °C/0.28 at 8 °C) and salmon fillets (0.29 at 5 °C/0.41 at 8 °C), indicating that sushi salmon offers the most favorable conditions for growth of *L. monocytogenes* in this setup (Figure 1, *p*-values for the difference between the slope of the growth curve of sushi salmon compared to that of Norwegian smoked salmon and salmon fillets: *p* = 0.084 at 5 °C, *p* = 0.048 at 8 °C).

In Norwegian smoked salmon injected with low-NaL brine, growth of *L. monocytogenes* was observed between t = 0 and t = 16. During this time, *L. monocytogenes* increased by 1.7 log CFU/g at 5 °C and 3.5 log CFU/g at 8 °C, respectively. At 5 °C, a marked decrease in *L. monocytogenes* numbers was observed at t = 15 with a return at t = 16 to numbers that were more in agreement with the slope of the curve at earlier time points.

In salmon fillets, the largest increase in *L. monocytogenes* log CFU/g was detected, even though the shelf life and therefore also the incubation in the laboratory was one day shorter than for the Norwegian smoked salmon. Between t = 0 and t = 15, *L. monocytogenes* increased by 2.7 log CFU/g at 5 °C and 3.6 log CFU/g at 8 °C, respectively.

### 3.3. Growth Potential δ

According to EURL Lm Technical Guidance document [26], a product will be classified as “supports the growth of *L. monocytogenes*” or “does not support the growth of *L. monocytogenes*” based on the growth potential δ. We therefore calculated the growth potential according to the guideline for each salmon product (Table 3). According to this guideline, all products were classified as “supports the growth of *L. monocytogenes*” (δ ≥ 0.5 log CFU/g), except Norwegian smoked salmon with low NaL kept at 5 °C for five days. Salmon fillets showed the highest δ among all tested products (4.95 log CFU/g at t = 14 at 8 °C).

### 3.4. Durability Study

After enrichment, *L. monocytogenes* was detected in at least three out of five samples of all tested products at all time points (Table 4). In most of the cases, the numbers of *L. monocytogenes* were below 1 log CFU/g. Quantitative detection of *L. monocytogenes* without enrichment was only possible in sushi salmon, in which *L. monocytogenes* exceeded 1 log CFU/g. The highest value was reported in sushi salmon after 8 days at eight degrees with 1.8 log CFU/g.

### 3.5. Total Viable Count

Initial TVCs (total viable counts) ranged between 2.4 ± 1.2 and 2.8 ± 0.2 log CFU/g (Figure 2). While sushi salmon and Norwegian smoked salmon showed relatively uniform TVC counts, these numbers had a larger range in the salmon fillets with the corresponding larger standard deviation. The highest TVC count among all tested samples was found after incubation at 8 °C in salmon fillets at day 13 (7.1 ± 1.3 log CFU/g,) and Norwegian smoked salmon at day 14 (7.1 ± 0.8 log CFU/g). There was a positive correlation between growth of *L. monocytogenes* and the growth of the total viable counts as determined by a linear regression (with an average *R*^2^ of 0.68).

### 3.6. Protective Effect of 30% NaL Injection into Norwegian Smoked Salmon

The protective effect of increasing the NaL concentration in the injected brine from 12 to 30% was tested for Norwegian smoked salmon (Figure 3).

#### 3.6.1. Growth Characteristics of *L. monocytogenes* in Norwegian Smoked Salmon Injected with High Concentration of NaL

In Norwegian smoked salmon injected with the high concentration of NaL, no growth of *L. monocytogenes* was observed at 5 °C between t = 0 and t = 14 (Figure 3a), with a slight increase of 0.57 ± 0.22 log CFU/g during the last two days of shelf life. At 8 °C, the maximum increase was 1.98 ± 0.68 log CFU/g. Compared to the Norwegian smoked salmon injected with low-NaL, the injection of high-NaL significantly reduced the growth of *L. monocytogenes* in Norwegian smoked salmon by an average rate of 11% at 5 °C (*p* = 0.001) and 6% at 8 °C (*p* < 0.001).

#### 3.6.2. Growth Potential δ

After assessing growth behavior of *L. monocytogenes* in the salmon injected with high-NaL, the growth potential δ was calculated (Table 5). For the Norwegian smoked salmon injected with high-NaL and stored at 5 °C, δ exceeded 0.5 log CFU/g only at t = 15. At 8 °C, δ was higher than 0.5 log CFU/g at all sample times except for t = 15 (0.4 log CFU/g).

#### 3.6.3. Total Viable Counts in Norwegian Smoked Salmon Injected with 30% NaL

The injection of NaL at the high concentration had an inhibitory effect on the TVC counts in Norwegian smoked salmon at both temperatures (Figure 3b). Compared to the same product injected with low-NaL, TVC were reduced by 12% at 5 °C (*p* = 0.09) and by 14% at 8 °C (*p* = 0.02).

## 4. Discussion

The results from this study add to the evidence that salmon is a high-risk product for growth of *L. monocytogenes*. Even at the storage temperature of 5 °C, the growth potential exceeded 0.5 log CFU/g and therefore put the product in the category “able to support the growth of *L. monocytogenes*” (EC 2073/2005). Generally, we observed a rather large variability in our data. We followed the “EURL Lm Technical Guidance Document for conducting shelf-life studies on *Listeria monocytogenes* in ready-to-eat food” [26], which is the current standard for European Union Reference Laboratories. However, due to the measurement uncertainty in quantifying bacteria, some criticism has been raised at its accuracy to reliably discriminate a growth potential of <1 log CFU with an α of 0.05 and a power of 0.80 [29].

In line with another study [10], sushi salmon was confirmed to be the product with the highest risk, justifying its short shelf life. In sushi salmon, the growth curves for both *L. monocytogenes* and TVC displayed the steepest slopes. In addition, salmon of sushi quality was the only product in which natural contamination levels of *L. monocytogenes* exceeding 1 log CFU/g were detected in the durability study.

While the challenge test demonstrated a growth potential of *L. monocytogenes* in Norwegian smoked salmon (low-NaL) and salmon fillets in the range of 2.21 to 4.70 log CFU/g at the final time points of the experiments, there was no observable growth in the durability studies in these products, as determined by direct plating. However, up to 100% of the products used in the durability study were contaminated with *L. monocytogenes* at low levels. We would therefore have expected to see growth of *L. monocytogenes* at comparable levels to the challenge test. This discrepancy may be due to microbial injury leading to prolonged lag times, or due to interactions with the competitive flora of the salmon products [30,31,32]. In challenge tests, *L. monocytogenes* might be the predominant microflora in spiked products, therefore having a competitive advantage over other microorganisms, whereas in naturally contaminated foods, this advantage may be lost as *L. monocytogenes* levels are often very low.

The addition of NaL negatively affected the growth potential of *L. monocytogenes*. Norwegian smoked salmon, even when injected with the lower concentration of NaL at 12%, was the only product that was not classified as “supports the growth of *L. monocytogenes*” at all time points.

The addition of NaL at the higher concentration of 30% seemed to efficiently reduce the growth potential of *L. monocytogenes* in Norwegian smoked salmon, which confirms results that have shown NaL to decrease the proliferation of spoilage organisms in refrigerated sliced salmon [25]. The effect of high-NaL injection on the growth of *L. monocytogenes* in Norwegian smoked salmon at 5 °C was large enough to change its classification to “does not support the growth of *L. monocytogenes*” according to EC regulation No 2073/2005. The classification of products according to this regulation is based on the growth potential at the end of the challenge test, with recommended, but not mandatory additional time points. Even though we adhered to the extensive number of plates required to determine *L. monocytogenes* growth by the EC 2073/2005 regulation, there was considerable variation in the data. This emphasizes the importance of analyzing additional time points to mitigate inherent variability in natural products and the fact that the growth potential could peak at some other time point before the end of the shelf life. As an example, at t = 15, the growth potential of *L. monocytogenes* in Norwegian smoked salmon injected with high-NaL was 1.1 log CFU/g and largely exceeded the limit of 0.5 log CFU/g, while at the end of the shelf life it did not. These are important considerations when interpreting growth data from challenge tests. The effect of NaL has been attributed to lowering the pH and reducing the water activity [33]. We did not observe changes in the physicochemical properties of the Norwegian smoked salmon injected with the high concentration of NaL compared to those in Norwegian smoked salmon injected with the low concentration of NaL (Table 2). The pH (6.0) stayed the same; and the minimal changes in a_w_ values (0.95–0.96) would not be expected to impact *L. monocytogenes* growth. In agreement with other studies [34], it is plausible that the undissociated form of NaL exerts an antimicrobial effect on the growth of *L. monocytogenes* that is independent from changes in pH and a_w_ values. The water activity of salmon also depends on the curing method. Wet-cured products like the salmon fillets and the Norwegian smoked salmon used in this study were injected with a brine solution, resulting in higher a_w_ values (a_w_: 0.962–0.964) (this study, [35,36]), whereas dry-curing, where salt and/or sugar was applied directly onto the surface, resulted in slightly lower a_w_ values of 0.931–0.948 [37]. NaL has also been shown to have a beneficial effect in packaging films and protective coatings: films that incorporated NaL in different combinations with either nisin or sodium diacetate prevented the growth of *L. monocytogenes* on cold smoked salmon [38,39].

Cold, salt, acid and alkali stress, including NaL can cause filamentation (“chaining”) in *L. monocytogenes* due to incomplete cell separation during growth [40,41]. The possibility of chaining should be considered when interpreting observations of reduced growth of *L. monocytogenes* in salmon after the addition of NaL, because this phenomenon would lead to an underestimation of bacterial counts by direct plating since a single chain of many cells will result in only one colony.

In conclusion, the salmon products analyzed here must be considered high-risk products due to their intrinsic properties. The addition of NaL to salmon products may increase food safety by lowering the growth potential of *L. monocytogenes*. However, it should be viewed as a complementation to good manufacturing practices and cannot compensate a lack thereof. This also emphasizes the uninterrupted maintenance of the cold chain as an important contribution to food safety.

## Figures and Tables

**Figure 1 foods-09-01048-f001:**
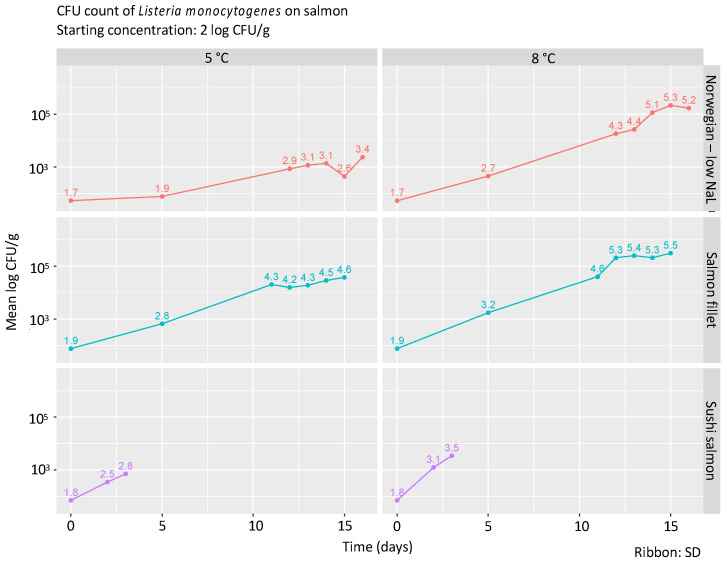
Growth curves of *L. monocytogenes*. The numbers above the time points reflect the mean log CFU/g from three replicates. The shaded area around the line represents the standard deviation.

**Figure 2 foods-09-01048-f002:**
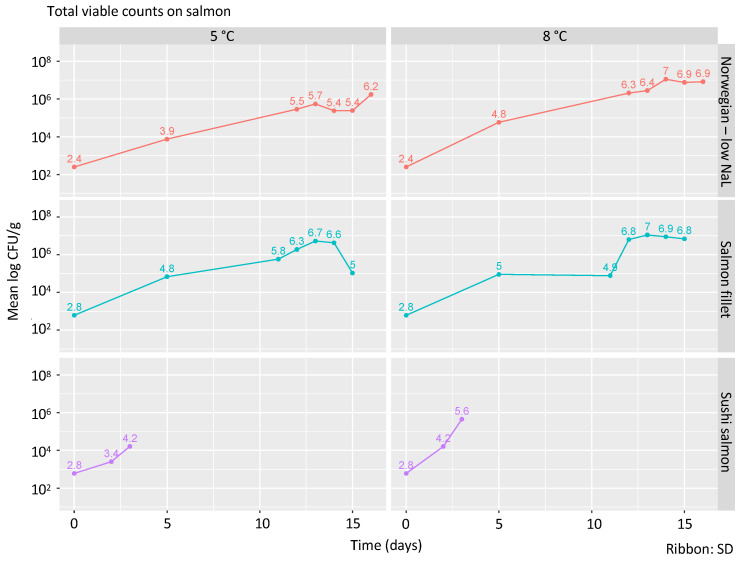
Total count of viable bacteria at 37 °C. The numbers above the time points reflect the mean log CFU/g from three replicates. The shaded area around the line represents the standard deviation.

**Figure 3 foods-09-01048-f003:**
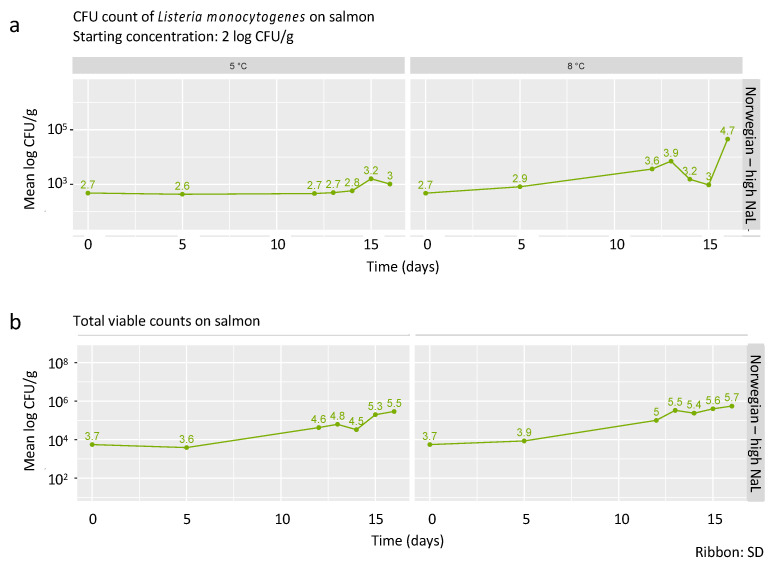
Growth curves of *L. monocytogenes* (**a**) and TVC (**b**) on Norwegian smoked salmon injected with 30% NaL. The numbers above the time points reflect the mean log CFU/g from three replicates. The shaded area around the line represents the standard deviation.

**Table 1 foods-09-01048-t001:** *L. monocytogenes* strains used in this study.

Bacterial Species	Designation	Serotype	CgMLST ^1^	Sequence Type ^2^
*Listeria monocytogenes*	N18-1945	1/2a	5048	121
*Listeria monocytogenes*	N18-2495	1/2a	2812	204
*Listeria monocytogenes*	N18-2497	1/2b	1271	5

^1^ core genome MLST. ^2^ as determined by seven-gene MLST.

**Table 2 foods-09-01048-t002:** Salmon products (Salmo salar) used in this study.

Salmon Product	Description	Preservation Method (If Any)	Storage	Packaging Units	Shelf Life	a_w_—Value (SD ^1^)	pH (SD ^1^)
Norwegian smoked salmon (low concentration of NaL)	Injection-salted smoked Atlantic salmon from Norway	Brine-injected Working solution of 12% NaL and 20% NaCl, 100–120 g/kg.	Cooled to −3 °C immediately after processing	200 g	16 days	0.96 (0.002)	6.0 (0.0)
Salmon fillet	Hand-salted smoked organic Atlantic salmon fillet from Ireland	Salted Sea salt (NaCl) manually added to the outside of the product; 30 g/kg	Cooled to −3 °C immediately after processing	110 g	15 days	0.97 (0.006)	6.0 (0.1)
Sushi salmon	Raw Atlantic sushi salmon from Scotland	No conservation Raw, trimmed salmon	Shock-frozen after packaging	whole fillets	3 days	0.99 (0.003)	6.1 (0.1)
Norwegian smoked salmon (with high concentration of NaL)	Injection-salted, smoked and sliced Atlantic salmon from Norway	Brine-injected Working solution of 30% NaL and 13% NaCl, 100–120 g/kg.	Cooled to −3 °C immediately after processing	200 g	16 days	0.95 (0.003)	6.0 (0.1)

^1^ SD: standard deviation.

**Table 3 foods-09-01048-t003:** Growth potential δ of *L. monocytogenes* in salmon products according to Commission Regulation (EC) No 2073/2005.

Salmon Product	°C	t ^1^	Growth Potential δ Replicate 1 ^2^	Growth Potential δ Replicate 2 ^2^	Growth Potential δ Replicate 3 ^2^	Max. Growth Pot. δ (Italic If δ > 0.5) ^2, 3^
Sushi salmon	5 °C	0	0.00	0.00	0.00	0.00
	5 °C	2	0.67	0.70	0.63	*0.70*
	5 °C	3	0.87	0.95	1.05	*1.05*
	8 °C	0	0.00	0.00	0.00	0.00
	8 °C	2	1.20	1.09	1.27	*1.27*
	8 °C	3	1.16	1.87	1.98	*1.98*
Norwegian smoked salmon (low NaL)	5 °C	0	0.00	0.00	0.00	0.00
	5 °C	5	0.21	0.15	0.22	0.22
	5 °C	12	1.94	1.73	0.47	*1.94*
	5 °C	13	1.45	1.09	0.64	*1.45*
	5 °C	14	1.68	0.40	2.14	*2.14*
	5 °C	15	1.18	0.96	0.93	*1.18*
	5 °C	16	2.21	1.82	0.69	*2.21*
	8 °C	0	0.00	0.00	0.00	0.00
	8 °C	5	0.79	0.72	0.93	*0.93*
	8°C	12	2.84	2.80	2.07	*2.84*
	8 °C	13	3.31	2.97	2.30	*3.31*
	8 °C	14	3.46	3.95	3.19	*3.95*
	8 °C	15	n/a	3.34	3.72	*3.72*
	8 °C	16	3.57	3.47	2.87	*3.57*
Salmon fillet	5 °C	0	0.00	0.00	0.00	0.00
	5 °C	5	1.24	1.25	0.47	*1.25*
	5 °C	11	2.45	2.78	1.76	*2.78*
	5 °C	12	2.44	2.63	1.72	*2.63*
	5 °C	13	2.26	2.43	2.28	*2.43*
	5 °C	14	2.48	2.73	2.63	*2.73*
	5 °C	15	2.78	3.78	1.30	*3.78*
	8 °C	0	0.00	0.00	0.00	0.00
	8 °C	5	2.05	1.58	0.33	*2.05*
	8 °C	11	3.61	2.88	2.08	*3.61*
	8 °C	12	3.62	3.23	3.10	*3.62*
	8 °C	13	3.53	3.37	3.29	*3.53*
	8 °C	14	3.70	4.95	1.99	*4.95*
	8 °C	15	2.83	4.70	3.33	*4.70*

^1^ time in days. ^2^ in log CFU/g. ^3^ Italic: value is larger than 0.5, putting the product in the category “able to support the growth of. *L. monocytogenes*” (EC 2073/2005).

**Table 4 foods-09-01048-t004:** Durability study with *L. monocytogenes* on different salmon products.

z	Day	Temperature	Proportion Positive after Enrichment ^1^	CI ^2^	Quantitative Determination by Direct Plating ^3^
Salmon fillet	0	n/a	0.6	0.22–0.88	<0
	11	4 °C	0.8	0.35–0.95	<0
	11	8 °C	1	0.54–0.99	<0
	13	4 °C	1	0.54–0.99	<0
	13	8 °C	0.8	0.35–0.95	<0
	14	4 °C	1	0.54–0.99	<0
	14	8 °C	0.8	0.35–0.95	<0
Norwegian smoked salmon	0	n/a	1	0.54–0.99	<0
	13	4 °C	1	0.54–0.99	<0
	13	8 °C	0.8	0.35–0.95	<0
	15	4 °C	1	0.54–0.99	<0
	15	8 °C	1	0.54–0.99	<0
	16	4 °C	1	0.54–0.99	<0
	16	8 °C	1	0.54–0.99	<0
Sushi salmon	0	n/a	1	0.54–0.99	<0
	2	4 °C	1	0.54–0.99	<0
	2	8 °C	1	0.54–0.99	<0
	3	4 °C	1	0.54–0.99	1
	3	8 °C	1	0.54–0.99	1.7
	6	4 °C	1	0.54–0.99	<0
	6	8 °C	1	0.54–0.99	1
	8	4 °C	0.6	0.22–0.88	<0
	8	8 °C	0.8	0.35–0.95	1.8
	10	4 °C	1	0.54–0.99	<0
	10	8 °C	1	0.54–0.99	1.5

^1^*n* = 5. ^2^ CI: confidence interval. ^3^ in log CFU/g. Limit of detection: 1 log CFU/g. The highest value among 5 replicates is reported.

**Table 5 foods-09-01048-t005:** Growth potential δ of *L. monocytogenes* in Norwegian smoked salmon injected with 30% NaL, according to Commission Regulation (EC) No 2073/2005.

Salmon Product	°C	t ^1^	Growth Potential δ Replicate 1 ^2^	Growth Potential δ Replicate 2 ^2^	Growth Potential δ Replicate 3 ^2^	Max. Growth Pot. δ (Italic If δ > 0.5) ^2, 3^
Norwegian smoked salmon	5 °C	0	0.00	0.00	0.00	0.00
(high NaL)	5 °C	5	0.02	−0.16	0.02	0.02
	5 °C	12	0.00	−0.03	−0.09	0.00
	5 °C	13	−0.06	0.08	−0.09	0.08
	5 °C	14	−0.11	0.09	0.07	0.09
	5 °C	15	0.05	1.01	0.80	*1.01*
	5 °C	16	−0.09	0.24	0.32	0.32
	8 °C	0	0.00	0.00	0.00	0.00
	8 °C	5	0.06	−0.03	0.62	*0.62*
	8 °C	12	0.28	1.11	0.23	*1.11*
	8 °C	13	1.51	1.19	0.76	*1.51*
	8 °C	14	0.49	0.02	1.20	*1.20*
	8 °C	15	0.42	−0.11	0.11	0.42
	8 °C	16	1.19	2.12	2.30	*2.30*

^1^ time in days. ^2^ in log CFU/g. ^3^ Italic: value is larger than 0.5, putting the product in the category “able to support the growth of. *L. monocytogenes*” (EC 2073/2005).

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
