# Peer review of "Growth Potential of Listeria monocytogenes in Three Different Salmon Products"

_foods, 2020, doi:10.3390/foods9081048_

Round 1

Reviewer 1 Report

The stated purpose of the paper was to compare the growth potential of Listeria monocytogenes in three salmon RTE products, by using a challenge test and a durability study, followed by the evaluation of the effects of brining with sodium lactate on Norwegian smoked salmon.

The manuscript is interesting and expands the knowledge on the subject investigated. The paper is well organized, concise, and easy to read. The English language used is fine but some misprints are present in the text.

In my opinion, all the figures and the tables are necessary to explain the results obtained.

However, to improve the quality of the study, I would recommend the changes described below.

Specific comments:

Abstract

L19-20: I would say: “to prevent growth” instead of “to prevent contamination with high numbers”.

Introduction

L68: for salmon, “dry-salted” would be probably better than “salt-rubbed”.

Materials and methods:

L77: The authors stated how they modified the method described in the European guidelines (EURL Lm Technical Guidance Document). As these guidelines require that one of the strains should have known growth characteristics, it would be important to provide more details on the isolation of the strains used (e.g. isolated from raw materials or smoked salmon or environment, etc.), and state in the first paragraph (L74-82) if the growth characteristics of all the strains were determined before inoculation or at the same time, as described in the EURL guidelines (page 17) for the challenge strain control.

L99-100: Please state the medium used.

Table 2: “Storage” should be capitalized.

L105-115: Were the samples of the four salmon products obtained from the same manufacturer? Please indicate their origin. Please indicate the species: Salmo salar? Did the manufacturer carry out any phosphate treatment on the samples?

Results

L199 and L218: “fillet” instead of “filet”.

Figure 1 and Figure 2: I cannot see the ribbon showing the standard deviation.

L208: “salmon fillet” not capitalized.

L263: L. monocytogenes in italics.

Discussion

L284: I wouldn’t consider the temperature of 5°C “legally mandated”, as in most of the countries a temperature range between 0°C and 4°C is recommended. Here I would write: “even at the storage temperature of 5°C”.

L291: L. monocytogenes in italics.

L296-297: The authors should discuss the differences observed between low Na-L brined samples and dry-salted samples. In this respect, the salting method might have played a role in determining a higher growth potential in dry-salted samples. Serio et al. (Food Control 22, 2011, 2071-2075) already observed that L.m. strains isolated from dry-salted salmon have a unique electrophoretic profile and a distinct behaviour.

L300-L307: Among the different causes of the discrepancies between the results obtained in the challenge test and in the durability test, the authors might wish to mention the intrinsic factors regarding the limited power and the errors described for the EURL method by Powell (International Journal of Food Microbiology 136, 2009, 10-17). In this paper, Powell concluded that EURL method cannot provide estimates with power and error that can be considered suitable for regulatory purposes, and suggested that it should be necessary to increase the number of samples or reduce uncertainty.

Author Response

Reviewer 1:
The stated purpose of the paper was to compare the growth potential of Listeria monocytogenes in three salmon RTE products, by using a challenge test and a durability study, followed by the evaluation of the effects of brining with sodium lactate on Norwegian smoked salmon.
The manuscript is interesting and expands the knowledge on the subject investigated. The paper is well organized, concise, and easy to read. The English language used is fine but some misprints are present in the text.
In my opinion, all the figures and the tables are necessary to explain the results obtained.
However, to improve the quality of the study, I would recommend the changes described below.
Specific comments:
Abstract
L19-20: I would say: “to prevent growth” instead of “to prevent contamination with high numbers”.
This has been changed.
Introduction
L68: for salmon, “dry-salted” would be probably better than “salt-rubbed”.
This has been changed.
Materials and methods:
L77: The authors stated how they modified the method described in the European guidelines (EURL Lm Technical Guidance Document). As these guidelines require that one of the strains should have known growth characteristics, it would be important to provide more details on the isolation of the strains used (e.g. isolated from raw materials or smoked salmon or environment, etc.), and state in the first paragraph (L74-82) if the growth characteristics of all the strains were determined before inoculation or at the same time, as described in the EURL guidelines (page 17) for the challenge strain control.
I have added that the growth curves were performed under the same storage conditions as the challenge test. These growth curves were done prior to the challenge tests, I don’t think adding this information will add value to the description?
Unfortunately, the information on what product exactly these strains were originally isolated from is not available. All we know is that they have been isolated in the framework of self-control from a salmon producing facility, which is why we deemded them relevant strains.
L99-100: Please state the medium used.
Good point.
Table 2: “Storage” should be capitalized.
This has been changed.
L105-115: Were the samples of the four salmon products obtained from the same manufacturer? Please indicate their origin. Please indicate the species: Salmo salar? Did the manufacturer carry out any phosphate treatment on the samples?
This information has been added. The origin is specified in table 2.
Results
L199 and L218: “fillet” instead of “filet”.
This has been changed.

Figure 1 and Figure 2: I cannot see the ribbon showing the standard deviation.
Maybe “ribbon” was not a good word choice. I have replaced it by “shaded area”. Does it make sense now?
L208: “salmon fillet” not capitalized.
This has been changed.
L263: L. monocytogenes in italics.
This has been changed.
Discussion
L284: I wouldn’t consider the temperature of 5°C “legally mandated”, as in most of the countries a temperature range between 0°C and 4°C is recommended. Here I would write: “even at the storage temperature of 5°C”.
This has been changed.
L291: L. monocytogenes in italics.
This has been changed.
L296-297: The authors should discuss the differences observed between low Na-L brined samples and dry-salted samples. In this respect, the salting method might have played a role in determining a higher growth potential in dry-salted samples. Serio et al. (Food Control 22, 2011, 2071-2075) already observed that L.m. strains isolated from dry-salted salmon have a unique electrophoretic profile and a distinct behaviour.
I am not convinced that Serio et al. did not isolate the same strain several times.
L300-L307: Among the different causes of the discrepancies between the results obtained in the challenge test and in the durability test, the authors might wish to mention the intrinsic factors regarding the limited power and the errors described for the EURL method by Powell (International Journal of Food Microbiology 136, 2009, 10-17). In this paper, Powell concluded that EURL method cannot provide estimates with power and error that can be considered suitable for regulatory purposes, and suggested that it should be necessary to increase the number of samples or reduce uncertainty.
This reference has been added.

Reviewer 2 Report

Ms:  Growth potential of Listeria monocytogenes in three different salmon products

The present manuscript describes the potential growth of L. monocyotgenes in some types of fish products.

The introduction section is clear and the experimental carried out is well described but not properly organized. However the ms presented some points to be addressed:

  • Ms described the use o f salmon of sushi quality. It seems other fish products presented lower quality. I think better you may indicate sample as “sushi salmon” or salmon intended for sushi preparation or something like that
  • The material and methods section may improve if divided in some parts such as: inoculum preparation, sample preparation and challenge test.
  • The microbiological counts along the ms are not present in the same units. It is recommended to present it as Log CFU/g (including tables)
  • The Results should be divided according to the type of food. It has no sense compared the growth of monocytogenes in fresh and smoked salmon as indicated in line 208
  • Some paragraphs of results section are difficult to understand
  • The ms should be formatted properly to avoid some mistakes such as table 1, L. monocytogenes in italic, units across the text, etc…
  • Discussion is not properly addressed and presented several errors. For example, in line indicated “several studies” but only one reference.
  • In line 291, authors indicated that growth of monocytogenes is higher in fresh salmon than smoked salmon (obviously!) and then indicated “In fact, doubling times of L. monocytogenes in fresh salmon close to those in brain 293 heart infusion have been reported “ (line 293) What is the meaning of this sentence?
  • Also, discussion presented in paragraphs from lines 287 to 307 and from 337 to 344 are confusing and not really related with the work

In overall, the ms presented a proper introduction and a well designed experimental part. However, the current work is not novelty and only addresses the growth of L. monocytogenes in selected fish fresh products. The results are not properly presented and the discussion is not really clear at all. Results and discussion should be revised accordingly.

To helps authors, some points have been indicated along the manuscript.

INTRODUCTION

Line 25: In particular the elderly, pregnants, chidrens and immunosuppressed people are at increased risk for listeriosis. Please change pregnant women by pregants (it is understand that pregnants are women),

Line 31: change "refrigerator temperature" by cold storage

Line 33-35: You repeat the same concept described in line 33,

Line 57 and 64: Remove the annex. Only refer (Regulation 2073/2004) as reference

Line 65-72: this paragraph belongs to material and method section

MATERIAL AND METHODS

The material and method section should be divided in subsection such as i) preparation of the inoculum, ii) salmon preparation, iii) challenge test,

The microorganisms counts should be presented as log CFU/g since authors present the results in this format,

Lines 105-109: The information presented is described in table 2. You should indicate something like that in the text "the characteristics of salmon samples used in the present work are presented in table 2",

Line 131: Please put the concentration values in number,

Line 147: Please change "test units" by samples throughout the text,

Line 148-150: please write the equipment characteristics in brackets,

Line 159: please add the correct symbol for microlitres,

Lines 168-170: you repeat the same information described in lines 144-146

RESULTS

Line 189: please change "rich medium" by non-selective growth media, "properties" by "characteristics and "test units" by samples

Line 196: You tested the growth of L. monocytogenes in different salmon products. I think the salmon used in the work is the same for all the different products and belongs to the Salmo salar specie. Is that correct?,

Line 197: as previously indicated, please describe the microbiological counts as Log CFU/g,

Line 200: please write the number followed by +/- symbol and then the SD (ex: 1,45+0,32),

Lines 237-241. please present the microbiological counts as log CFU/g,

In table 4, what is the importance of the confidence interval in the proportion of positive samples?

Lines 255-258: this paragraph is difficult to understand,

Lines 260-262: this paragraph is difficult to understand

Table 1 is not properly formatted

DISCUSSION

Line 289: please, change durabilty to "shelf life",

Line 291. L. monocytogenes in italic,

Lines 287-307- this paragraph is really difficult to understand,

Line 341 How do you identificate L.monocytogenes throughout by staining?

Author Response

Reviewer 2:

The present manuscript describes the potential growth of L. monocyotgenes in some types of fish products.

The introduction section is clear and the experimental carried out is well described but not properly organized. However the ms presented some points to be addressed:

  • Ms described the use o f salmon of sushi quality. It seems other fish products presented lower quality. I think better you may indicate sample as “sushi salmon” or salmon intended for sushi preparation or something like that
    This has been changed, we are now calling it “sushi salmon”.
  • The material and methods section may improve if divided in some parts such as: inoculum preparation, sample preparation and challenge test.
    The journal does not allow for subtitles in the M&M section.
  • The microbiological counts along the ms are not present in the same units. It is recommended to present it as Log CFU/g (including tables)
    Originally we opted for the CFU/g representation for the results of the durability study because we feared that the notion of “0 log CFU/g” to represent 10 CFU/g might be confusing to read. But we agree that it is more consistent this way and have changed it to be log CFU/g throughout the manuscript.
  • The Results should be divided according to the type of food. It has no sense compared the growth of monocytogenes in fresh and smoked salmon as indicated in line 208
    The results section has been reorganized according to the suggestions of Reviewer 3. I agree that it needed reorganizing.
  • Some paragraphs of results section are difficult to understand
    see above comment.
  • The ms should be formatted properly to avoid some mistakes such as table 1, L. monocytogenes in italic, units across the text, etc…
    This has been fixed.
  • Discussion is not properly addressed and presented several errors. For example, in line indicated “several studies” but only one reference.
  • This has been fixed.
  • In line 291, authors indicated that growth of monocytogenesis higher in fresh salmon than smoked salmon (obviously!) and then indicated “In fact, doubling times of  monocytogenes in fresh salmon close to those in brain 293 heart infusion have been reported “ (line 293) What is the meaning of this sentence?
    This sentence has been deleted.  
  • Also, discussion presented in paragraphs from lines 287 to 307 and from 337 to 344 are confusing and not really related with the work
    These parts of the discussion has been reworked.

In overall, the ms presented a proper introduction and a well designed experimental part. However, the current work is not novelty and only addresses the growth of L. monocytogenes in selected fish fresh products. The results are not properly presented and the discussion is not really clear at all. Results and discussion should be revised accordingly.

They have both been revised.

To helps authors, some points have been indicated along the manuscript.

We have revised the results and discussion according to your comments and those of the other reviewers.

INTRODUCTION

Line 25: In particular the elderly, pregnants, chidrens and immunosuppressed people are at increased risk for listeriosis. Please change pregnant women by pregants (it is understand that pregnants are women),

done

Line 31: change "refrigerator temperature" by cold storage

done

Line 33-35: You repeat the same concept described in line 33,

this has been shortened.

Line 57 and 64: Remove the annex. Only refer (Regulation 2073/2004) as reference

done

Line 65-72: this paragraph belongs to material and method section

Agreed.

MATERIAL AND METHODS

The material and method section should be divided in subsection such as i) preparation of the inoculum, ii) salmon preparation, iii) challenge test,

The word template that MDPI mandates to be used has no subsections for the methods and materials section.

The microorganisms counts should be presented as log CFU/g since authors present the results in this format,

this has been changed, good point.

Lines 105-109: The information presented is described in table 2. You should indicate something like that in the text "the characteristics of salmon samples used in the present work are presented in table 2",

This has been done.

Line 131: Please put the concentration values in number,

As one is not supposed to start a sentence with a numner, I consider this the accepted workaround.

Line 147: Please change "test units" by samples throughout the text,

This has been done.

Line 148-150: please write the equipment characteristics in brackets,

This has been done.

Line 159: please add the correct symbol for microlitres,

This has been done.

Lines 168-170: you repeat the same information described in lines 144-146

Lines 168-170 gives the time points for the durability study, lines 144-146 those for the challenge test. They differ slightly, the durability test for sushi salmon was longer than the challenge test. Mainly because we were curious what would happen and we were keeping  the other varieties that long anyway.

RESULTS

Line 189: please change "rich medium" by non-selective growth media, "properties" by "characteristics and "test units" by samples

This has been done.

Line 196: You tested the growth of L. monocytogenes in different salmon products. I think the salmon used in the work is the same for all the different products and belongs to the Salmo salar specie. Is that correct?,

Yes, this is correct and has been added accordingly.

Line 197: as previously indicated, please describe the microbiological counts as Log CFU/g,

This has been done.

Line 200: please write the number followed by +/- symbol and then the SD (ex: 1,45+0,32),

This has been done.

Lines 237-241. please present the microbiological counts as log CFU/g,

This has been done.

In table 4, what is the importance of the confidence interval in the proportion of positive samples?

This means if you test this many samples and get a proportion of 0.6, the true value is between 0.22-0.88, p<0.05

Lines 255-258: this paragraph is difficult to understand,

Agreed. It has been freed of unnecessary bulk.

Lines 260-262: this paragraph is difficult to understand

It has been changed.

Table 1 is not properly formatted

This has been changed.

DISCUSSION

Line 289: please, change durabilty to "shelf life",

Since the “EURL Lm technical guidance document” calls them “durability studies”, I would prefer to stick with durability.

Line 291. L. monocytogenes in italic,

This has been changed, the sentence has been deleted anyway.

Lines 287-307- this paragraph is really difficult to understand,

It has been changed.

Line 341 How do you identificate L.monocytogenes throughout by staining?

This bit has been deleted. But to answer your question: the experiment was done on a pure culture of L. mono. Therefore, anything that stained in a Gram stain was presumed to be L. mono.

Reviewer 3 Report

I have reviewed the manuscript foods-854712 titled “Growth potential of Listeria monocytogenes in three different salmon products”. Salmon is a highly valuable food product from a global perspective and salmon raised in aquaculture has become commonplace in many countries where this product is a huge economic driver. Unfortunately, salmon products have often been implicated in large numbers of food recalls due to the presence of L. monocytogenes. Therefore, authors’ aim in the present study to determine the risk posed by this pathogen in these products is substantiated. The study was essentially in two parts: First, following the recommended EURL technical guidelines to determine the growth potential of L. monocytogenes, a three strain cocktail of Lm was used to determine the growth of the pathogen in sushi quality salmon, salmon fillets and Norwegian smoked salmon containing 12 % sodium lactate (NaL) at two different temperatures (the recommended refrigerated temp. and an abusive temp.). Secondly, the authors aimed to improve the safety of Norwegian smoked salmon, by increasing the NaL levels and again following the EURL technical guidelines to ascertain the growth potential of Listeria. Somehow I think this should also be reflected in the title.

Overall the study was carried out properly in adherence to the aforementioned guidelines and the interpretation of data was correct. The Introduction section was well written and the Methods were adequately explained. However, my main concern with the manuscript was the presentation of the Results. This is mainly due to the choice of merging data from the two separate studies. As the results were written it became confusing to see low and high NaL production in the same Figures and Tables since the high concentration product was not mentioned until the very end of the Results section, within its own sub-section. I strongly feel these results need to be separated because the goal of the first set of experiments was in fact to compare Lm growth in three commercially available salmon products side by side. Then, based on these observed results you ran challenges to improve the safety for one of these products (i.e. Norwegian smoked salmon). The way you have written the text portion of the Results section as well as the Discussion begs for separate Tables and Figures for the high NaL smoked salmon as this is a product improvement endeavor. For example add a Figure 3 with Lm and TVC counts and a Table 5 for the Growth Potential of Lm om the high NaL.

My specific comments and suggested detailed edits are listed below:

Line 13: Change to “…on all of the tested salmon products, …”

Lines 16-18: You said “The effect of the injection of sodium lactate (NaL) at a high concentration (30 %) into cold smoked salmon was also determined.” without following it up by telling us what was the outcome. I think a line needs to be added after this sentence.

Line 27: Delete parenthesis before “which” and place a comma after 2017.

Line 29: Change to “…infections of the central nervous system.”

Line 30: Is it proper to start a sentence with an abbreviated Genus name? I am old school so I would say not. Please spell out Listeria or rearrange sentence so it does begin with L. monocytogenes.

Line 43: 5-6 log CFU per g?

Line 44: Change to “These events demonstrate…”

Line 49: Should spell out the US Food and drug Administration

Line 49: Change live to life

Line 51: Rewrite sentence “There is no consensus on acceptable levels of L. monocytogenes in food.”

Line 53: Shelf life

Line 56: should this be “… <100 CFU/g in five 25 g samples?

Line 60: FBO not defined

Line 89: In the next step…

Line 91: Delete “overnight” as it seems redundant.

Line 109: “Salmon fillets were hand-salted…”

Line 110: “…smoked salmon and salmon fillets were…”

Line 112: “salmon fillets”

Lines 124-125: Did you mean “…and to one unit not inoculated with L. monocytogenes, NaCl was added for aw and pH measurements.”?

Line 125: 10 g

Lines 126-127: “For each salmon variety, one additional non-inoculated test unit was prepared to detect natural contamination with L. monocytogenes.” Would you expect this level of sampling to be adequate to detect “natural” Lm contamination?

Line 130: Shouldn’t the target concentration for Lm inoculated on the product in a challenge study actually be below the allowable limit (i.e. 100 CFU/g)? In Canada for example, a level of 10-30 CFU/g is used and levels should not exceed 100 CFU/g at any point during the stated shelf life of the product.

Lines 132-136: “The inoculated and control samples, including the carton carrier, were placed in polyamide-polyethylene vacuum bags (Solis vacuum packaging bags no 922.61, Glattbrugg, Switzerland) with a pair of sterile tweezers, the bags were vacuumized and sealed with a commercially available vacuum sealer (Solis Easyvac Pro 569, Glattbrugg, Switzerland).” I assume this is how these products would be packaged and sold commercially?

Line 137: Replace whole with entire

Line 141: “Salmon fillets were analyzed…”

Line 145: salmon fillets

Line 157: Change grade to speed

Line 157: Space missing for “100ul” use proper symbol for µl

Line 174: Space missing 48h

Line 178: Should be “Data quality analysis and outlier detection was performed with Knime…”

Line 194: The “w” for water activity abbreviation should be subscript.

Lines 197-198: You stated “..significant growth of L. monocytogenes was observed in all salmon products…”, however, looking at Figure 1 there appears to be relatively little growth (<0.5 log CFU/g) over the 16 days of incubation at 5°C.

Line 199: salmon fillets

Line 200: I rarely see standard deviations presented like this. Would it make more sense to present as “…with 4.6 ± 1.14 log CFU/ g at 5 °C and 5.5 ± 0.75 log CFU/g at 8 °C, respectively.”?

The caption for Figure 1 on Page 7 (BTW the page numbers shown at the top right of each page are incorrect) states “The ribbon around the line represents the standard deviation.” However, I do not see a “ribbon” around any of the lines. Why not simply use error bars to show the STDEV?

Lines 206-208: How was the slope of the growth curves calculated. Was this a simply linear regression line through all values for all sample days or was a more complex model applied?

Line 208: Do not capitalize “salmon fillet”

Line 208: Here, I assume this statement means you are referring to only the Norwegian smoked salmon with low NaL? This needs to be clarified.

Line 218:” In salmon fillets” or you could say “filleted salmon”

Line 222: Change “without” to “excluding”

Line 222: Also, I am not sure what you mean by (without the Norwegian smoked salmon with low-NaL). Did you mean (…with high NaL)?

Line 231: According to Table 3 on Day 15 the growth potential in the high NaL product stored at 5°C exceeded 0.5 log CFU/g (1.01 CFU/g), yet on Day 16 it was below this limit. How can one account for this variability when trying to designate a product as “not supporting the growth of Lm)? Do you believe this anomaly was due to a technical error?

Table 3: I really hate the layout of this Table! Do you really need the product names and temperatures to be repeated down the entire column. I makes the Table look far too busy. I have included an example of a nicer layout.

Table 4: Again can you simply group the Product name so not to repeat on every line. Also in the last column of the headings row, Quantitative should be capitalized and I don’t think you need to say “of L. monocytogenes” because we know from the Table title we are talking about L. monocytogenes; hence “Quantitative determination by direct plating” will suffice.

Line 244: As suggested above I would simply go with 2.4 ± ? log CFU/g instead of placing the STDEV values in parentheses?

Lines 244-245: By this sentence are you referring to the TVC range on all salmon products tested? This needs to be stated.

Line 247: Should be 8°C

Line 248: Same issue with the display of STDEV

Figure 2 Caption: “The ribbon around the line represents the standard deviation.” Not visible in the document I received. Again why not simply use error bars?

Lines 259-262: This seems out of place, especially since the data is in Table 2 presented much earlier. Does this really need a separate sub-section or can this simply be injected into the Discussion. I think the latter.

Line 263: Italicize L. monocytogenes

Line 267: “maximum”

Lines 267-268: Again the STDEV thing.

Line 269: Delete extra space between salmon and by.

Line 275: “…at all sample times except…”

Line 275: In all earlier instances in the manuscript you used log CFU/g and now you specify log10 CFU/g. Please be consistent and use the same format throughout.

Line 280: Why is this Table template here? This is very sloppy.

Line 287: Change to “…risk, as the growth curves for both L. monocytogenes and TVC on this product displayed the steepest slopes. In addition, salmon of sushi quality was the only product in which natural contamination levels of L. monocytogenes exceeding 10 CFU/g were detected in the durability study.”

Line 291: Italicize L. monocytogenes

Line 292: Rewrite as “…and faster growth of L. monocytogenes was observed in fresh salmon…”

Line 293: change to “BHI broth”

Line 297: Choose either log CFU/g or log10 CFU/g and stick with it.

Line 298: “…there was no observable growth…” as determined by direct plating?

Lines 316-318: You wrote “The classification of products according to this regulation is based on the growth potential at the end of the challenge test, with recommended, but not mandatory additional time points.” Why does the regulation specify the end of the shelf life. Isn’t it possible that growth potential could peak at some other time point before the end? Or am I misinterpreting you?

Line 320: I suppose this answers me previous questions. J

Line 332 and 333: aw values does not need to be hyphenated

Line 343: Is the virulence potential of filamented L. monocytogenes known? Do filamented morphotypes revert to normal cells once ingested.

RECOMMENDED FORMAT FOR TABLE 3 (Perhaps also center justify all cells)

Salmon Product

Temp.

°C

Time

(Days)

Growth Potential (δ)

Replicate

Maximum δ

1

2

3

Salmon of sushi quality

5°C

0

0.00

0.00

0.00

0.00

2

0.67

0.70

0.63

0.70

3

0.87

0.95

1.07

1.07

8°C

0

0.00

0.00

0.00

0.00

2

1.20

1.09

1.27

1.27

3

1.16

1.87

1.98

1.98

Norwegian smoked salmon

(low NaL)

5°C

0

0.00

0.00

0.00

0.00

5

0.21

0.15

0.22

0.22

12

1.94

1.73

0.47

1.94

13

1.45

1.09

0.64

1.45

14

1.68

0.40

2.14

2.14

15

1.18

0.96

0.93

1.18

16

2.21

1.82

0.69

2.21

8°C

0

0.00

0.00

0.00

0.00

5

0.79

0.72

0.93

0.93

12

2.84

2.80

2.07

2.84

13

3.31

2.97

2.30

3.31

14

3.46

3.95

3.19

3.95

15

n/a

3.34

3.72

3.72

16

3.57

3.47

2.87

3.57

Author Response

Please refer to the attached file for a more pleasant and readable version of the answers. 

Reviewer 3:

I have reviewed the manuscript foods-854712 titled “Growth potential of Listeria monocytogenes in three different salmon products”. Salmon is a highly valuable food product from a global perspective and salmon raised in aquaculture has become commonplace in many countries where this product is a huge economic driver. Unfortunately, salmon products have often been implicated in large numbers of food recalls due to the presence of L. monocytogenes. Therefore, authors’ aim in the present study to determine the risk posed by this pathogen in these products is substantiated. The study was essentially in two parts: First, following the recommended EURL technical guidelines to determine the growth potential of L. monocytogenes, a three strain cocktail of Lm was used to determine the growth of the pathogen in sushi quality salmon, salmon fillets and Norwegian smoked salmon containing 12 % sodium lactate (NaL) at two different temperatures (the recommended refrigerated temp. and an abusive temp.). Secondly, the authors aimed to improve the safety of Norwegian smoked salmon, by increasing the NaL levels and again following the EURL technical guidelines to ascertain the growth potential of Listeria. Somehow I think this should also be reflected in the title.

I am playing around with different versions of the title. “Growth potential of Listeria monocytogenes in three different salmon products and effect of different NaL concentrations”. “Effect of different NaL concentrations on the growth potential of Listeria monocytogenes in three different salmon products”. “Risk assessment for L. monocytogenes in three different salmon products”. I like none of them better than the original, and they feel bulky, so I am choosing not to change the title.

Overall the study was carried out properly in adherence to the aforementioned guidelines and the interpretation of data was correct. The Introduction section was well written and the Methods were adequately explained. However, my main concern with the manuscript was the presentation of the Results. This is mainly due to the choice of merging data from the two separate studies. As the results were written it became confusing to see low and high NaL production in the same Figures and Tables since the high concentration product was not mentioned until the very end of the Results section, within its own sub-section. I strongly feel these results need to be separated because the goal of the first set of experiments was in fact to compare Lm growth in three commercially available salmon products side by side. Then, based on these observed results you ran challenges to improve the safety for one of these products (i.e. Norwegian smoked salmon). The way you have written the text portion of the Results section as well as the Discussion begs for separate Tables and Figures for the high NaL smoked salmon as this is a product improvement endeavor. For example add a Figure 3 with Lm and TVC counts and a Table 5 for the Growth Potential of Lm om the high NaL.

This has been done.

My specific comments and suggested detailed edits are listed below:

Line 13: Change to “…on all of the tested salmon products, …”

This has been done.

Lines 16-18: You said “The effect of the injection of sodium lactate (NaL) at a high concentration (30 %) into cold smoked salmon was also determined.” without following it up by telling us what was the outcome. I think a line needs to be added after this sentence.

Agreed.

Line 27: Delete parenthesis before “which” and place a comma after 2017.

This has been done.

Line 29: Change to “…infections of the central nervous system.”

This has been done.

Line 30: Is it proper to start a sentence with an abbreviated Genus name? I am old school so I would say not. Please spell out Listeria or rearrange sentence so it does begin with L. monocytogenes.

I did not know that – learned something new.

Line 43: 5-6 log CFU per g?

This has been done.

Line 44: Change to “These events demonstrate…”

This has been done.

Line 49: Should spell out the US Food and drug Administration

This has been done.

Line 49: Change live to life

This has been done.

Line 51: Rewrite sentence “There is no consensus on acceptable levels of L. monocytogenes in food.”

This has been done.

Line 53: Shelf life

This has been done.

Line 56: should this be “… <100 CFU/g in five 25 g samples?

No, this is correct:

Line 60: FBO not defined

This has been changed.

Line 89: In the next step…

This has been changed.

Line 91: Delete “overnight” as it seems redundant.

This has been done.

Line 109: “Salmon fillets were hand-salted…”

This sentence has been deleted.

Line 110: “…smoked salmon and salmon fillets were…”

This sentence has been deleted.

Line 112: “salmon fillets”

This has been changed.

Lines 124-125: Did you mean “…and to one unit not inoculated with L. monocytogenes, NaCl was added for aw and pH measurements.”?

Yes.

Line 125: 10 g

This has been changed.

Lines 126-127: “For each salmon variety, one additional non-inoculated test unit was prepared to detect natural contamination with L. monocytogenes.” Would you expect this level of sampling to be adequate to detect “natural” Lm contamination?

No. Per product/time/temperature combination, there were three samples inoculated with L. monocytogenes, and three negative controls inoculated with NaCl. If the salmon was naturally contaminated with L. monocytogenes, we would have had a decent chance to detect it in these negative samples. The additional “non-inoculated test unit at t=0” was sort of a double negative control: not even NaCl was added to them. The point of this was in case all our negative controls (those to which NaCl had been added) showed up positive, this double negative control would show us whether the NaCl had been contaminated with L. monocytogenes. It wasn’t, obviously.

Line 130: Shouldn’t the target concentration for Lm inoculated on the product in a challenge study actually be below the allowable limit (i.e. 100 CFU/g)? In Canada for example, a level of 10-30 CFU/g is used and levels should not exceed 100 CFU/g at any point during the stated shelf life of the product.

Since this study was done adhering to the EU technical guidance document for challenge tests with L. monocytogenes, we used the target concentration of 100 CFU/g stated there. Adhering to the guideline was also necessary because the food producer used the data for his documentation.

Lines 132-136: “The inoculated and control samples, including the carton carrier, were placed in polyamide-polyethylene vacuum bags (Solis vacuum packaging bags no 922.61, Glattbrugg, Switzerland) with a pair of sterile tweezers, the bags were vacuumized and sealed with a commercially available vacuum sealer (Solis Easyvac Pro 569, Glattbrugg, Switzerland).” I assume this is how these products would be packaged and sold commercially?

Correct.

Line 137: Replace whole with entire

This has been changed.

Line 141: “Salmon fillets were analyzed…”

This has been changed.

Line 145: salmon fillets

This has been changed.

Line 157: Change grade to speed

This has been changed.

Line 157: Space missing for “100ul” use proper symbol for µl

This has been changed.

Line 174: Space missing 48h

This has been changed.

Line 178: Should be “Data quality analysis and outlier detection was performed with Knime…”

This has been changed.

Line 194: The “w” for water activity abbreviation should be subscript.

This has been changed.

Lines 197-198: You stated “..significant growth of L. monocytogenes was observed in all salmon products…”, however, looking at Figure 1 there appears to be relatively little growth (<0.5 log CFU/g) over the 16 days of incubation at 5°C.

I meant the increase was statistically significant, the sentence has been changed accordingly.

Line 199: salmon fillets

This has been changed.

Line 200: I rarely see standard deviations presented like this. Would it make more sense to present as “…with 4.6 ± 1.14 log CFU/ g at 5 °C and 5.5 ± 0.75 log CFU/g at 8 °C, respectively.”?

This has been changed.

The caption for Figure 1 on Page 7 (BTW the page numbers shown at the top right of each page are incorrect) states “The ribbon around the line represents the standard deviation.” However, I do not see a “ribbon” around any of the lines. Why not simply use error bars to show the STDEV?

Maybe «ribbon» was not a good word choice. I have replaced it wiht «shaded area». Does it make sense now?

Lines 206-208: How was the slope of the growth curves calculated. Was this a simply linear regression line through all values for all sample days or was a more complex model applied?

The regression line was fitted to the exponential part of the curve only. 

Line 208: Do not capitalize “salmon fillet”

This has been changed.

Line 208: Here, I assume this statement means you are referring to only the Norwegian smoked salmon with low NaL? This needs to be clarified.

Yes. «low NaL» has been added.

Line 218:” In salmon fillets” or you could say “filleted salmon”

This has been changed.

Line 222: Change “without” to “excluding”

This has been changed.

Line 222: Also, I am not sure what you mean by (without the Norwegian smoked salmon with low-NaL). Did you mean (…with high NaL)?

No. In the linear regression model, the p-value for time as a predictor of growth of Norwegian smoked salmon with low NaL was slightly above 0.05, therefore not significant.

Line 231: According to Table 3 on Day 15 the growth potential in the high NaL product stored at 5°C exceeded 0.5 log CFU/g (1.01 CFU/g), yet on Day 16 it was below this limit. How can one account for this variability when trying to designate a product as “not supporting the growth of Lm)? Do you believe this anomaly was due to a technical error?

No. I believe there are several factors at work here: the inherent measurement uncertainty of plate counts, differences in the pieces of salmon, differences in the flora on individual pieces. The student who performed the work is not generally incapable of reproducing reliable plate counts: when she plated the inoculum, she was able to hit the target concentration with an accuracy of around 0.1 log CFU/ml, which I deem very acceptable. Overall, this highlights that it is a good idea to add additional time points to the mandatory t=0 and t=end stated in the EU technical guidance.

Table 3: I really hate the layout of this Table! Do you really need the product names and temperatures to be repeated down the entire column. I makes the Table look far too busy. I have included an example of a nicer layout.

I’m very sorry you hated it so much! And thanks for the suggested layout, which I’m using now. It does look nicer now.

Table 4: Again can you simply group the Product name so not to repeat on every line. Also in the last column of the headings row, Quantitative should be capitalized and I don’t think you need to say “of L. monocytogenes” because we know from the Table title we are talking about L. monocytogenes; hence “Quantitative determination by direct plating” will suffice.

This has been changed.

Line 244: As suggested above I would simply go with 2.4 ± ? log CFU/g instead of placing the STDEV values in parentheses?

This has been changed.

Lines 244-245: By this sentence are you referring to the TVC range on all salmon products tested? This needs to be stated.

It is now stating this.

Line 247: Should be 8°C

This has been changed.

Line 248: Same issue with the display of STDEV

This has been changed.

Figure 2 Caption: “The ribbon around the line represents the standard deviation.” Not visible in the document I received. Again why not simply use error bars?

See comment to figure 1.

Lines 259-262: This seems out of place, especially since the data is in Table 2 presented much earlier. Does this really need a separate sub-section or can this simply be injected into the Discussion. I think the latter.

I like the idea.

Line 263: Italicize L. monocytogenes

This has been changed.

Line 267: “maximum”

This has been changed.

Lines 267-268: Again the STDEV thing.

This has been changed.

Line 269: Delete extra space between salmon and by.

This has been changed.

Line 275: “…at all sample times except…”

This has been changed.

Line 275: In all earlier instances in the manuscript you used log CFU/g and now you specify log10 CFU/g. Please be consistent and use the same format throughout.

This has been changed.

Line 280: Why is this Table template here? This is very sloppy.

Yes. Very.

Line 287: Change to “…risk, as the growth curves for both L. monocytogenesand TVC on this product displayed the steepest slopes. In addition, salmon of sushi quality was the only product in which natural contamination levels of L. monocytogenes exceeding 10 CFU/g were detected in the durability study.”

This has been changed.

Line 291: Italicize L. monocytogenes

This sentence has been deleted.

Line 292: Rewrite as “…and faster growth of L. monocytogenes was observed in fresh salmon…”

This sentence has been deleted.

Line 293: change to “BHI broth”

This sentence has been deleted.

Line 297: Choose either log CFU/g or log10 CFU/g and stick with it.

This has been changed.

Line 298: “…there was no observable growth…” as determined by direct plating?

Yes, has been added.

Lines 316-318: You wrote “The classification of products according to this regulation is based on the growth potential at the end of the challenge test, with recommended, but not mandatory additional time points.” Why does the regulation specify the end of the shelf life. Isn’t it possible that growth potential could peak at some other time point before the end? Or am I misinterpreting you?

No, you are neither misinterpreting me nor the regulation. I agree that this is not optimal. I could perform a challenge test that has t=0 and t=end and if I detect growth of L. monocytogenes <0.5 log CFU between the two points, the testing criteria for “product does not support growth of L. monocytogenes” will apply. I stole your sentence “that growth potential could peak at some other time point before the end» and added it to the discussion.

Line 320: I suppose this answers me previous questions. J

Line 332 and 333: aw values does not need to be hyphenated

This has been changed.

Line 343: Is the virulence potential of filamented L. monocytogenes known? Do filamented morphotypes revert to normal cells once ingested.

I have decided to delete this part. It was a quick and dirty way to check whether they filament after the addition of NaL, but has too many question marks. We did the filamentation experiments in BHI, do the bacteria behave the same on salmon with NaL? What are the optimal conditions to test this? Shaking, not shaking, what temperature? We tested a few conditions and never saw filaments, which in the end means nothing, really.  

RECOMMENDED FORMAT FOR TABLE 3 (Perhaps also center justify all cells)

Salmon Product

Temp.

°C

Time

(Days)

Growth Potential (δ)

Replicate

Maximum δ

1

2

3

Salmon of sushi quality

5°C

0

0.00

0.00

0.00

0.00

2

0.67

0.70

0.63

0.70

3

0.87

0.95

1.07

1.07

8°C

0

0.00

0.00

0.00

0.00

2

1.20

1.09

1.27

1.27

3

1.16

1.87

1.98

1.98

Norwegian smoked salmon

(low NaL)

5°C

0

0.00

0.00

0.00

0.00

5

0.21

0.15

0.22

0.22

12

1.94

1.73

0.47

1.94

13

1.45

1.09

0.64

1.45

14

1.68

0.40

2.14

2.14

15

1.18

0.96

0.93

1.18

16

2.21

1.82

0.69

2.21

8°C

0

0.00

0.00

0.00

0.00

5

0.79

0.72

0.93

0.93

12

2.84

2.80

2.07

2.84

13

3.31

2.97

2.30

3.31

14

3.46

3.95

3.19

3.95

15

n/a

3.34

3.72

3.72

16

3.57

3.47

2.87

3.57

Round 2

Reviewer 2 Report

The ms have been improved as requested.

Some minor details must to be addressed.

  • Line 19: L. monocytogenes in italics
  • Keywords: L. monocytogenes in italic
  • Line 27: I think this information should be eliminated “EU member states reported 2480 confirmed cases in 2017,which corresponds to an incidence of 270.48 cases/100’000 population”. The reference is enough
  • Line 35:  I think this information is not necessary since you indicated the importance of the disease in line 28 “with the severity of the disease with mortality rates between 15-30 35%”
  • Line 61: You stated “high aW”. If you refer to RTE meat products such as cured/fermented meat products (as the recent outbreak in Spain 3 months ago), these products are characterized by both low pH and aW
  • Line 122: Change the units into Log CFU/g
  • Line 142: “Salmon product was refrigerated at -20ºC”. Refrigerated or frozen at -20ºC
  • Table 2: Please add “Table 2.Salmon products (Salmo salar) used in this study
  • The font size of the information of Figure 1 is really small and difficult to read.
  • Line 377: 8 degrees (in number)
  • Line 380: please change +/- by the specific word symbol ±
  • Figure 3: font size, as indicated in figure 2, is really small
  • Line 601: laboratories
  • Line 612-614: This sentence is difficult to understand.
  • Lines 616-618: this sentence is not clear at all
  • Line 621: “….was the only product that not “supports the growth of L. monocytogenes“ in all sampling time
  • Line 640: values
  • Lines 718-720: How do authors justify the hypothesis of chaining?  

Author Response

Reviewer 2 (2nd review)

The ms have been improved as requested. Some minor details must to be addressed.

Line 19: L. monocytogenes in italics

This has been changed.

Keywords: L. monocytogenes in italic

This has been changed.

Line 27: I think this information should be eliminated “EU member states reported

2480 confirmed cases in 2017,which corresponds to an incidence of 270.48

cases/100’000 population”. The reference is enough

This has been changed.

Line 35: I think this information is not necessary since you indicated the

importance of the disease in line 28 “with the severity of the disease with mortality

rates between 15-30 35%”

Line 61: You stated “high aW”. If you refer to RTE meat products such as

cured/fermented meat products (as the recent outbreak in Spain 3 months ago), these

products are characterized by both low pH and aW

The references refer to outbreaks with seafood or meat pate. Since this is given as an example, I don’t feel like it negates the possibility that outbreaks can also be due to products with lower aw-values.

Line 122: Change the units into Log CFU/g

This has been changed.

Line 142: “Salmon product was refrigerated at -20oC”. Refrigerated or frozen at -

20oC

This has been changed to “frozen”

Table 2: Please add “Table 2.Salmon products (Salmo salar) used in this study

This has been changed.

The font size of the information of Figure 1 is really small and difficult to read.

I agree. This is mainly due to the fact that I have pasted the figure into the .docx document for your reference, this is not meant to be the final format. It should be fixed in typesetting.

Line 377: 8 degrees (in number)

This has been changed.

line 380: please change +/- by the specific word symbol ±

This has been changed.

Figure 3: font size, as indicated in figure 2, is really small

See comment above for figure 1.

Line 601: laboratories

This has been changed.

Line 612-614: This sentence is difficult to understand.

This sentence has been rewritten.

Lines 616-618: this sentence is not clear at all

This sentence has been rewritten according to reviewer 3’s suggestion.  

Line 621: “....was the only product that not “supports the growth of L.

monocytogenes“ in all sampling time

This sentence has been rewritten.

Line 640: values

The values from Table 2 have been added to this text passage.

Lines 718-720: How do authors justify the hypothesis of chaining?

It has been shown that cold, salt and pH stress, may induce filamentation/chaining due to incomplete cell separation. I have clarified the sentence to make clear that filamentation and chaining are the same thing.

Reviewer 3 Report

I am satisfied with the revisions the authors have made to the manuscript. My only other edits are of a minor nature and are listed below.

Line 67-68: Change to “…EU regulations stipulate a food safety criterion of < 100 CFU/g during the shelf life of the product (Commission Regulation (EC) No 2073/2005).”

Line 139: Capitalize Table 2

Line 142: Should be “…frozen at -20°C…”

Line 198: This sentence should not be a new paragraph

Figure 1 – In the original PDF that I received to review, there was no “ribbon” or shaded area around the time points. Now seeing this version makes sense to me.

Line 270: Should be (Supplementary Figure S1)

Line 319: salmon fillets

Line 383: salmon fillets

Lines 563-564: You switched your convention for log values. Suddenly you have the base 10 subscript. Please be consistent.

Line 601: Should be “Laboratories”

Line 616-618: Rewrite this sentence as “In challenge tests, L. monocytogenes might be the predominant microflora in spiked products, therefore having a competitive advantage over other microorganisms, whereas in naturally contaminated foods, this advantage may be lost as L. monocytogenes levels are often very low.

Line 664: salmon fillets

Line 720-721: To avoid redundancy of the word “chaining” in the same sentence rewrite as “…of NaL, because this phenomenon would lead to an underestimation of bacterial counts by direct plating since a single chain of many cells will result in only one colony.”

Author Response

Dear Reviewer 3, 

Please find a point-by-point answer to your review requests below. 

Line 67-68: Change to “…EU regulations stipulate a food safety criterion of < 100 CFU/g during the shelf life of the product (Commission Regulation (EC) No 2073/2005).”

This has been changed. 

Line 139: Capitalize Table 2

This has been changed. 

Line 142: Should be “…frozen at -20°C…”

This has been changed. 

Line 198: This sentence should not be a new paragraph

This has been changed.

Figure 1 – In the original PDF that I received to review, there was no “ribbon” or shaded area around the time points. Now seeing this version makes sense to me.

Ok I'm glad it worked out now. 

Line 270: Should be (Supplementary Figure S1)

Yep. 

Line 319: salmon fillets

I can't get my word to display line numbers that match with your comments, so I'm having to guess here. I did a search over the whole document and found a few instances where I agree it should be "fillets" plural. 

Line 383: salmon fillets

See above.

Lines 563-564: You switched your convention for log values. Suddenly you have the base 10 subscript. Please be consistent.

This has been changed. Found a few more and fixed those, too. 

Line 601: Should be “Laboratories”

This has been changed. 

Line 616-618: Rewrite this sentence as “In challenge tests, L. monocytogenes might be the predominant microflora in spiked products, therefore having a competitive advantage over other microorganisms, whereas in naturally contaminated foods, this advantage may be lost as L. monocytogenes levels are often very low.

That does read much less clunky, thanks. 

Line 664: salmon fillets

This has been changed. 

Line 720-721: To avoid redundancy of the word “chaining” in the same sentence rewrite as “…of NaL, because this phenomenon would lead to an underestimation of bacterial counts by direct plating since a single chain of many cells will result in only one colony.”

This has been changed.